

# Iron homeostasis in a mouse model of thalassemia intermedia is altered between adolescence and adulthood

Chanita Sanyear[1],[*], Punnee Butthep[1], Wiraya Eamsaard[2],
Suthat Fucharoen[3], Saovaros Svasti[3] and Patarabutr Masaratana[2],[*]

[1] Department of Pathology, Faculty of Medicine, Ramathibodi Hospital, Mahidol University, Bangkok, Thailand
[2] Department of Biochemistry, Faculty of Medicine Siriraj Hospital, Mahidol University, Bangkok, Thailand
[3] Thalassemia Research Center, Institute of Molecular Biosciences, Mahidol University, Bangkok, Thailand
[*] These authors contributed equally to this work.

Corresponding author
Patarabutr Masaratana,
patarabutr.mas@mahidol.ac.th

## ABSTRACT

**Background:** Iron overload is one of common complications of β-thalassemia. Systemic iron homeostasis is regulated by iron-regulatory hormone, hepcidin, which inhibits intestinal iron absorption and iron recycling by reticuloendothelial system. In addition, body iron status and requirement can be altered with age. In adolescence, iron requirement is increased due to blood volume expansion and growth spurt. Heterozygous β-globin knockout mice ($Hbb^{th3/+}$; BKO) is a mouse model of thalassemia widely used to study iron homeostasis under this pathological condition. However, effects of age on iron homeostasis, particularly the expression of genes involved in hemoglobin metabolism as well as erythroid regulators in the spleen, during adolescence have not been explored in this mouse model.

**Methods:** Iron parameters as well as the mRNA expression of hepcidin and genes involved in iron transport and metabolism in wildtype (WT) and BKO mice during adolescence (6–7 weeks old) and adulthood (16–20 weeks old) were analyzed and compared by 2-way ANOVA.

**Results:** The transition of adolescence to adulthood was associated with reductions in duodenal iron transporter mRNA expression and serum iron levels of both WT and BKO mice. Erythrocyte parameters in BKO mice remained abnormal in both age groups despite persistent induction of genes involved in hemoglobin metabolism in the spleen and progressively increased extramedullary erythropiesis. In BKO mice, adulthood was associated with increased liver hepcidin and ferroportin mRNA expression along with splenic erythroferrone mRNA suppression compared to adolescence.

**Conclusion:** Our results demonstrate that iron homeostasis in a mouse model of thalassemia intermedia is altered between adolescence and adulthood. The present study underscores the importance of the age of thalassemic mice in the study of molecular or pathophysiological changes under thalassemic condition.

## INTRODUCTION

Iron, one of essential trace elements, is involved in several biological processes such as oxidative phosphorylation and hemoglobin synthesis (*Jandl et al., 1959*; *Pollycove & Mortimer, 1961*; *Hentze, Muckenthaler & Andrews, 2004*). Iron is transported in the plasma by apotransferrin protein. Transferrin-bound iron is taken up into cells by transferrin receptor 1 (TfR1) for cellular storage or utilization. In erythroid cells, the acquired iron is used for the production of hemoglobin which involves several proteins including mitoferrin 1 (Mfrn1) a mitochondrial iron transporter and 5-aminolevulinic acid synthase 2 (ALAS2) an enzyme catalyzing the first reaction of erythroid heme biosynthetic pathway (*Shaw et al., 2006*; *Paradkar et al., 2009*; *Amigo et al., 2011*; *Lane et al., 2015*). Reticuloendothelial (RE) cells are responsible for the reutilization of iron in hemoglobin of senescent red blood cells. Upon erythrophagocytosis of senescent erythrocytes by RE cells, hemoglobin is degraded by heme oxygenase-1 (HO-1) and iron is released into the circulation for reutilization. This RE iron recycling process accounts for the majority of iron in the plasma. The minority of plasma iron is acquired from intestinal iron absorption (*Kong, Gao & Chang, 2014*; *Wallace, 2016*).

Several molecules are involved in the absorption of iron. Firstly, dietary iron ($Fe^{3+}$) is reduced by duodenal cytochrome b (Dcytb) located at the apical membrane of enterocytes (*McKie et al., 2001*). The resultant ferrous is taken up into enterocytes by divalent metal transporter 1 (DMT1) (*Gunshin et al., 1997*) and then transferred into the circulation by an iron efflux protein, ferroportin (*McKie et al., 2000*; *Donovan et al., 2000*; *Abboud & Haile, 2000*). The ferrous is subsequently oxidized by hephaestin into ferric which is then transported along the plasma by apotransferrin (*Vulpe et al., 1999*). Ferroportin not only involves in cellular export of dietary iron but also facilitates iron efflux from splenic macrophages and hepatocytes. In addition to intestinal iron uptake, DMT1 is also involved in the transport of transferrin-bound iron from endosome into cytoplasm.

Cellular iron homeostasis is regulated by iron responsive elements (IRE), which are located at either 5′ or 3′ untranslated region of several mRNAs encoding proteins related to cellular iron metabolism including TfR1, DMT1, ferroportin and ALAS2 (*Koeller et al., 1989*; *Cox et al., 1991*; *Dandekar et al., 1991*; *Abboud & Haile, 2000*; *McKie et al., 2000*; *Hubert & Hentze, 2002*). The expression of these proteins is post-transcriptionally controlled in response to cellular iron levels through the interaction between IRE and iron regulatory protein (IRP). TfR1 and DMT1 expression is increased under iron depletion state and vise versa under iron repletion. On the other hand, the expression of ferroportin and ALAS2 is suppressed in response to iron deficiency and enhanced during iron loading.

Systemic iron homeostasis is regulated by hepcidin, the liver-secreted iron regulatory hormone, which binds ferroportin and induces its internalization and degradation (*Nemeth et al., 2004*). Consequently, intestinal iron absorption and RE iron recycling is inhibited resulting in reduced serum iron levels. The expression of hepcidin is influenced by several factors. Body iron status and inflammation have been shown induce hepcidin expression whereas erythropoietic iron demand and hypoxia are hepcidin suppressors

(*Pigeon et al., 2001*; *Nicolas et al., 2002a*, *2002b*). Bone morphogenetic protein 6 (Bmp6) has been shown to play a crucial role in hepcidin induction in response to iron stores (*Andriopoulos et al., 2009*). In addition to the aforementioned factors, ineffective erythropoiesis can also suppress the expression of hepcidin. Growth differentiation factor 15 (Gdf15), twisted gastrulation 1 (Twsg1) and erythroferrone (Erfe) which are produced by erythroid precursors have been proposed to be potential candidates for hepcidin regulator under such conditions including thalassemia (*Tanno et al., 2007*, *2009*; *Kautz et al., 2014*).

Thalassemia is a hematological disease caused by mutations of globin-encoding genes or their promoters leading to decreased production of the respective globin chains. In β-thalassemia, the production of β-globin is reduced or absent leading to the accumulation of unmatched α-globin, erythroid cell death, ineffective erythropoiesis and, subsequently, anemia. Furthermore, systemic iron overload occurs as a result of increased intestinal iron absorption and/or regular blood transfusion with inadequate iron chelation (*Weatherall & Clegg, 2001*; *Higgs, Engel & Stamatoyannopoulos, 2012*). Animal models especially mouse (*Mus Musculus*) is the most widely used model to study the pathophysiological changes or treatment of human disease (*Capecchi, 2005*). Heterozygous β-globin knockout mice ($Hbb^{th3/+}$; BKO), one of commonly utilized thalassemia mouse model, were generated by heterozygous deletion of both murine β-globin genes (*Hbb-b1* and *Hbb-b2*) (*Detloff et al., 1994*; *Yang et al., 1995*). Phenotypic characterization of this mouse model includes mild-to-moderate anemia, growth retardation, ineffective erythropoiesis, extramedullary erythropoiesis and parenchymal iron loading in several tissues such as liver, spleen and heart which resemble clinical features of thalassemia in human (*Yang et al., 1995*).

Several factors including age, gender and strain have been shown to influence iron parameters as well as the expression of hepcidin and other iron regulatory genes (*Ahluwalia et al., 2000*; *Courselaud et al., 2004*; *Krijt et al., 2004*; *Weizer-Stern et al., 2006*; *Hahn et al., 2009*; *McLachlan et al., 2017*). A previous study reported that iron levels in various tissues were generally higher in aged mice than younger adult mice (*Hahn et al., 2009*). The expression of genes involved in iron metabolism was proposed to be dynamic as a result of age and iron stores (*De Franceschi et al., 2006*). Additionally, another previous study proposed that age must be considered in the studies of iron metabolism (*Chen et al., 2009*). It is noteworthy that mice at different ages might have distinctive responses to alterations in iron metabolism. Indeed, the study in β-thalassemia mouse model ($Hbb^{th3/+}$) aged 2, 5 and 12 months demonstrated different systemic iron regulatory responses in young adult and elderly thalassemic mice (*Gardenghi et al., 2007*). Although BKO mice have been widely used as an animal model in thalassemia research, little is known whether the expression of hepcidin along with major iron transporters in these mice differ across different age groups particularly in adolescence. Murine adolescent age is defined as the postnatal period ranging from weaning (PND 21) to adulthood (PND 60) (*Laviola et al., 2003*). It is also suggested that adult mice should be at least 3 months of age as most biological processes and structures continue to rapidly grow or mature after

the age of sexual maturation (35 days) until roughly 3 months old (*Flurkey, Currer & Harrison, 2007*). Moreover, iron status and iron homeostasis might be altered in adolescence as a result of increased body iron requirement for growth and development (*Beard, 2000*). Thus, the present study aims to explore the effects of age on iron parameters as well as the mRNA expression of hepcidin and major iron transport machineries in WT and thalassemic mouse model aged 6–7 weeks old and 16–20 weeks old, which represent adolescence and adulthood, respectively.

## MATERIALS AND METHODS

### Animal

Male C57BL/6 wild type (WT) and heterozygous β-globin knockout mice ($Hbb^{th3/+}$; BKO) at two age groups: 6–7 weeks old and 16–20 weeks old were used (five mice per each of the age group). All mice were obtained from the Thalassemia Research Center, Institute of Molecular Biosciences, Mahidol University, Thailand. The mice were given rodent chow (C.P. mice feed 082G containing 180 ppm of iron, Perfect Companion Group, Samut Prakan, Thailand) and water ad libitum. The temperature and humidity were maintained at 25 ± 2 °C and 55 ± 10%, respectively, with 12-h light/dark cycle. The mice were sacrificed and blood samples were collected by cardiac puncture. Liver, spleen and duodenum samples were snapped frozen and stored at −80 °C. Animal protocols were approved by Institute of Molecular Biosciences Animal Care and Use Committee, Mahidol University, Thailand (COA.NO.MB-ACUC 2016/003).

### Measurement of hematological and iron parameters

Hematological parameters were analyzed using an automated hematological analyzer (Mindray, Shenzhen, China). Serum iron level was determined using QuantiChrom iron assay kit (BioAssay Systems, Hayward, CA, USA) according to the manufacturer's protocol. Liver and spleen non-heme iron contents were determined by a modification of the method of *Foy et al. (1967)* as described by *Simpson & Peters (1990)*.

### Histopathological studies

Liver, spleen and duodenal tissue samples were fixed in 10% formalin buffer. The fixed tissue samples were then dehydrated, embedded in paraffin and sectioned at 4 μm thickness. Tissue sections were stained with hematoxylin and eosin (H&E) for morphological examination and Perl's Prussian blue for iron accumulation according to standard protocols. The stained slides were analyzed using a Nikon ECLIPSE 80i light microscope (Nikon, Shinagawa, Tokyo, Japan).

The amounts of mononuclear, polymorphonuclear and hematopoietic cells in the liver were evaluated in 10 non-overlapping areas acquired by random sampling. The numbers of cells were scored using a grading criteria described by *Yatmark et al. (2014)* as follows: score 0 = absent; score 1 = mild (1–10 cells); score 2 = moderate (11–50 cells); score 3 = severe (more than 50 cells).

For iron accumulation assessment, the whole tissue section of each mouse (five mice per group) was examined and the overall extent of iron deposition in each mouse was

estimated by a modification of grading system described by *Barton et al. (1995)* and *Yatmark et al. (2014)* as follows:

(a) Liver iron deposition: score 0 = no visible iron deposition; score 1 = slight iron deposition in the cytoplasm of Kupffer cells; score 2 = prominent iron accumulation in Kupffer cells; score 3 = iron deposition in hepatocytes; score 4 = iron deposition in hepatocytes and fibrous tissue of portal tracts or septa.

(b) Splenic iron deposition: score 0 = no visible iron deposition; score 1 = indicated iron deposits in marginal sinus of spleen; score 2 = densely aggregated iron deposition; score 3 = clumps of iron accumulation; score 4 = clumps of iron aggregation with frequent iron clumps.

(c) Duodenal iron deposition: score 0 = no visible iron deposition; score 1 = iron deposits in the supranuclear region of enterocyte; score 2 = densely aggregated iron deposition; score 3 = visible clumps of iron accumulation; 4 = clumps of iron aggregation with frequent iron clumps.

## Quantitative RT-PCR

RNA was extracted from the liver, spleen and duodenum using TRIzol reagent (Ambion, Austin, TX, USA) and complementary DNA was synthesized using a Tetro cDNA synthesis kit (Bioline USA, Taunton, MA, USA) as per manufacturers' protocols. Quantitative RT-PCR was performed using the CFX96 Thermal Cycler (Bio-Rad, Hercules, CA, USA). The gene expression was normalized to β-actin (*Actb*) expression. The results were presented as minus delta Ct values [$-(Ct_{target} - Ct_{Actb})$] (*Livak & Schmittgen, 2001*). The sequence of gene-specific primers is listed in Table 1.

## Statistical analysis

All data were expressed as mean ± standard error of the means (SEM). The comparison of the means across groups was performed by 2-way ANOVA with Bonferroni post-hoc test. A *P*-value less than 0.05 was considered significant. All statistical analyses were performed using GraphPad Prism 6 software (GraphPad Software, Inc., La Jolla, CA, USA).

# RESULTS

## Iron parameters, but not hematological parameters, of thalassemic mice were affected by age

To explore whether hematological parameters were affected by the age of mice, EDTA blood samples from WT and BKO mice aged 6–7 weeks and 16–20 weeks were analyzed for complete blood count. As shown in Table 2, significant effects of phenotype on several hematological parameters were observed. BKO mice had lower hemoglobin, hematocrit, MCV and RBC count along with higher RDW than age-matched WT which corresponded with thalassemia phenotype. Furthermore, such findings remained unaltered between adolescent and adult BKO mice. In WT mice, adulthood was associated with increased

**Table 1 Sequence of gene-specific primers.**

| Gene product | Forward primer | Reverse primer |
|---|---|---|
| *Actb* (β-actin) | CAGCCTTCCTTCTTGGGTA | TTTACGGATGTCAACGTCACAC |
| *Alas2* (ALAS2) | AGCCATTGTCCTTTCATGCT | CAGCAGGTCTGTCTTGAAAGTCT |
| *Bmp6* (Bmp6) | GCCAACTACTGTGATGGAGAGTGTT | CTCGGGATTCATAAGGTGGACCA |
| *Cybrd1* (Dcytb) | TTTGTCCTGAAACACCCCTC | AGAAGGCCCAGCGTATTTGT |
| *Fam132b* (Erfe) | TCCTCTATCTACAGGCAGGAC | ACTGCGTACCGTGAGGGA |
| *Gdf15* (Gdf15) | GAGCTACGGGGTCGCTTC | GGGACCCCAATCTCACCT |
| *Hamp* (Hepcidin) | CAGGGCAGACATTGCGATAC | GTGGCTCTAGGCTATGTTTTGC |
| *Hmox1* (HO-1) | CAGAGCCGTCTCGAGCATAG | CAAATCCTGGGGCATGCTGT |
| *Slc11a2* (DMT1) | TTCTACTTGGGTTGGCAGTGTT | CAGCAGGACTTTCGAGATGC |
| *Slc25a37* (Mfrn1) | ACGCCATGTATTTTGCCTGC | ACTCCCAGCTACCCCATTAG |
| *Slc40a1* (ferroportin) | ATCCCCATAGTCTCTGTCAGC | CAGCAACTGTGTCACCGTCA |
| *Tfrc* (TfR1) | TCCTTTCCTTGCATATTCTGG | CCAAATAAGGATAGTCTGCATCC |
| *Twsg1* (Twsg1) | GCTGTCACACCATGAAAACCTAG | ACTGTGCACATGCGCTCTT |

**Table 2 Hematological data and iron parameters of male wild type and thalassemic mice at the age of 6–7 weeks and 16–20 weeks.**

| | WT | | BKO | | P values (2-way ANOVA) | | |
|---|---|---|---|---|---|---|---|
| | 6–7 weeks | 16–20 weeks | 6–7 weeks | 16–20 weeks | Age | Phenotype | Age × phenotype |
| RBC count ($10^6$/μl) | 6.25 ± 0.45 | 8.91 ± 0.35[a] | 4.95 ± 0.45 | 4.68 ± 0.66[b] | 0.0270 | <0.0001 | 0.0087 |
| Hemoglobin (g/dL) | 9.84 ± 0.66 | 14.38 ± 0.65[a] | 6.46 ± 0.10[a] | 5.68 ± 0.90[b] | 0.0104 | <0.0001 | 0.0008 |
| Hematocrit (%) | 35.42 ± 1.98 | 44.66 ± 1.40[a] | 22.72 ± 1.70[a] | 19.18 ± 2.54[b] | 0.1634 | <0.0001 | 0.0048 |
| MCV (fL) | 57.06 ± 1.71 | 50.16 ± 0.47[a] | 46.38 ± 1.51[a] | 41.40 ± 1.87[b] | 0.0011 | <0.0001 | 0.5295 |
| MCH (pg) | 15.82 ± 0.65 | 16.12 ± 0.15 | 13.58 ± 1.52 | 12.24 ± 1.25[b] | 0.6235 | 0.0095 | 0.4415 |
| MCHC (g/dL) | 27.78 ± 1.10 | 32.14 ± 0.52 | 29.04 ± 2.24 | 29.34 ± 1.79 | 0.1541 | 0.6277 | 0.2109 |
| RDW (%) | 23.02 ± 1.62 | 12.88 ± 0.05[a] | 34.62 ± 2.69[a] | 27.28 ± 3.28[b] | 0.0014 | <0.0001 | 0.5462 |
| Serum iron (μM) | 53.21 ± 3.55 | 18.47 ± 1.69[a] | 44.44 ± 2.44 | 20.49 ± 3.27[c] | <0.0001 | 0.2509 | 0.0750 |
| Liver non-heme iron (nmole/mg wet weight) | 2.30 ± 0.34 | 1.92 ± 0.27 | 9.05 ± 1.67[a] | 8.72 ± 0.91[b] | 0.7207 | <0.0001 | 0.9799 |
| Spleen non-heme iron (nmole/mg wet weight) | 5.37 ± 0.40 | 7.50 ± 0.48 | 18.97 ± 1.33[a] | 59.22 ± 5.72[b,c] | <0.0001 | <0.0001 | <0.0001 |

Notes:
[a] *P*-value < 0.05 compared with WT aged 6–7 weeks.
[b] *P*-value < 0.05 compared with WT aged 16–20 weeks.
[c] *P*-value < 0.05 compared with BKO aged 6–7 weeks.
RBC, red blood cell; MCV, mean corpuscular volume; MCH, mean corpuscular hemoglobin; MCHC, mean corpuscular hemoglobin concentration; RDW, red cell distribution width.
Data are expressed as mean ± SEM (*n* = 5/group). Statistical analysis was performed by 2-way ANOVA with Bonferroni post-hoc test.

RBC count, hemoglobin, hematocrit along with reduced MCV and RDW compared to adolescence.

Serum iron levels as well as non-heme iron levels in the liver and spleen were measured to determine body iron status. Increased liver and spleen non-heme iron levels were observed in BKO mice whereas serum iron levels did not differ from WT mice (Table 2). In both WT and BKO mice, a significant reduction in serum iron levels were observed in

adult mice compared to adolescent mice while liver non-heme iron levels were unaltered. Interestingly, adulthood was associated with a significant increase in spleen non-heme iron levels only in BKO mice.

Perl's Prussian blue staining revealed iron accumulation in the liver and spleen of BKO mice (Figs. 1C–1D and 1G–1H). Notably, iron deposition was mainly confined in hepatic Kupffer cells and splenic macrophages in both red and white pulps. Additionally, a weakly positive iron staining was observed in the supranuclear region of duodenal enterocytes (Figs. 1K and 1L). The degree of iron staining was relatively higher in the spleen than the liver and duodenum (Table 3). In agreement with non-heme iron results, an increase in iron deposition score was found in the spleen of adult BKO group compared to adolescent BKO mice whereas liver and duodenal iron deposition score was unaltered between different age groups.

## The extent of extramedullary hematopoiesis in thalassemic mice was progressively increased with increasing age

Microscopic examination of liver and spleen tissue samples revealed hematopoietic and mononuclear cell infiltration in the liver and spleen of BKO mice which was suggestive for the presence of extramedullary hematopoiesis (Figs. 2C–2D and 2G–2H). As shown in Table 4, the livers of adult BKO mice had increased number of mononuclear cell infiltration and hematopoietic cells than adolescent BKO mice suggesting that the extent of extramedullary hematopoiesis progressively increased at least during the studied age range.

Furthermore, genes involved in hemoglobin synthesis (*Tfrc*, *Slc25a37* and *Alas2*) and degradation (*Hmox1*) were induced in the spleen of BKO mice compared to WT mice (Fig. 3). Interestingly, mRNA levels of these genes were significantly decreased in adult WT mice compared to adolescent counterpart while the expression in BKO mice was similar between the two age groups.

## Thalassemic mice had decreased liver hepcidin mRNA expression relative to liver non-heme iron contents during both adolescence and adulthood

Quantitative RT-PCR revealed that the levels of liver hepcidin mRNA were similar between WT and BKO mice during adolescence (Fig. 4A). In adulthood, hepcidin mRNA expression in BKO mice was higher than adolescent mice. On the contrary, hepcidin mRNA levels in WT were not affected by age. To account for liver iron overload, the mRNA expression of hepcidin was corrected by liver non-heme iron contents. As shown in Fig. 4B, BKO mice in both age groups had lower hepcidin mRNA levels relative to liver non-heme iron contents than WT mice. No effect of age on this parameter was observed.

The mRNA expression of upstream hepcidin regulators in the liver and spleen was also determined. Liver Bmp6 mRNA levels as well as splenic mRNA expression of Gdf15, Twsg1 and Erfe were higher in BKO mice compared to WT mice (Figs. 4C–4F). In general, the mRNA levels of these hepcidin regulators was not altered between adolescent and adult mice except splenic Erfe mRNA expression which was downregulated in adult BKO mice.

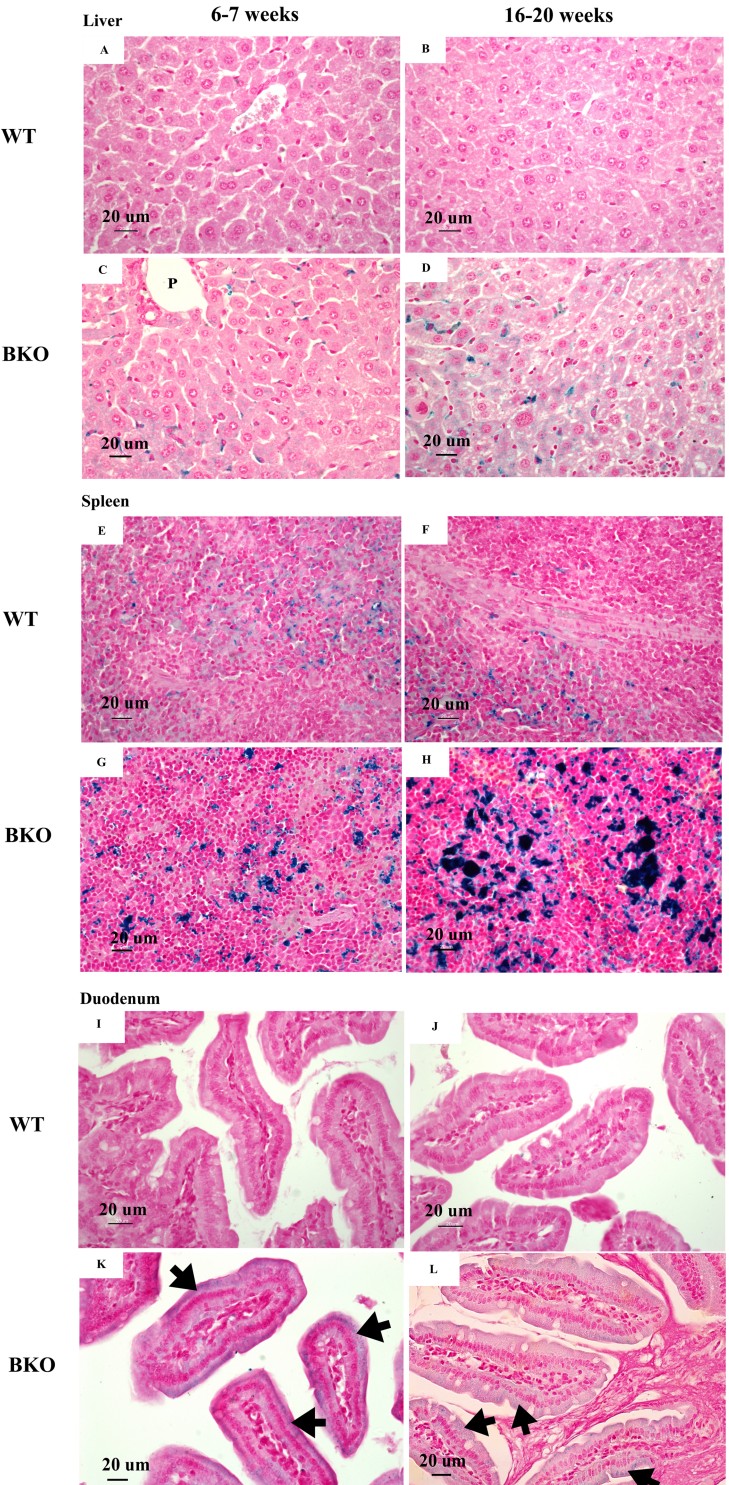

**Figure 1 Iron accumulation in the tissues of BKO mice.** Perl's Prussian blue staining of representative paraffin-embedded tissue sections from (A–D) the liver, (E–H) spleen and (I–L) duodenum of male WT and BKO mice aged 6–7 weeks and 16–20 weeks. The blue stain represents iron accumulation. The arrows indicate iron accumulation in brush border of enterocytes (Nikon ECLIPSE 80i light microscope; magnification ×400).

**Table 3  Iron deposition score in the liver, spleen and duodenum of male wild type and thalassemic mice at the age of 6–7 weeks and 16–20 weeks.**

| Age | Iron deposition score | | |
|---|---|---|---|
| | Liver | Spleen | Duodenum |
| WT | | | |
| 6–7 weeks | 0 | 1 | 0 |
| 16–20 weeks | 0 | 1 | 0 |
| BKO | | | |
| 6–7 weeks | 1 | 2 | 1 |
| 16–20 weeks | 1 | 3 | 1 |

**Note:**

The same deposition score was obtained from the mouse within same group ($n = 5$ per group).

## Iron transporter mRNA expression was differentially affected by thalassemia and age

In the liver, DMT1 mRNA expression was not affected by thalassemia and age (Fig. 5A). In contrast, ferroportin mRNA levels were increased in BKO mice (Fig. 5B). A significant ferroportin mRNA induction was observed in adult BKO mice compared to adolescent counterpart.

In addition, increased splenic mRNA levels of DMT1 and ferroportin were noted in BKO mice but the expression was not altered between adolescent and adult mice (Figs. 5C and 5D).

As shown in Fig. 6, thalassemia had significant positive effects of the mRNA expression of Dytb and DMT1 in the duodenum. Furthermore, Dcytb and DMT1 mRNA levels were significantly reduced in adult WT and BKO mice compared to respective adolescent group. On the other hand, duodenal ferroportin mRNA expression was not significantly affected by phenotype or age.

## DISCUSSION

Our study reveals that iron parameters and the expression of genes involved in iron homeostasis of WT and thalassemic mice differ between adolescence and adulthood. Although, previous study has determined hematological parameters and the expression of iron genes in thalassemic mice at different ages, the study focused on gene expression in the liver (De Franceschi et al., 2006). In addition, it was conducted only in mature adult and aged mice. The study by Gardenghi et al. (2007) utilized thalassemic mice aged 2, 5 and 12 months to compare iron status and iron homeostasis, however, the expression of splenic iron transporters as well as genes involved in hemoglobin synthesis and genes encoding erythriod regulators (Erfe, Gdf15, Twsg1) was not explored.

In accordance with previous studies (Jamsai et al., 2005; Nai et al., 2012; Upanan et al., 2015), we observed thalassemia intermedia phenotype in BKO mice during adolescence and adulthood as characterized by microcytic anemia, extramedullary hematopoiesis as well as iron accumulation in the liver and spleen. In the present study, hematological results revealed relatively lower red blood cell count and hemoglobin levels in both WT

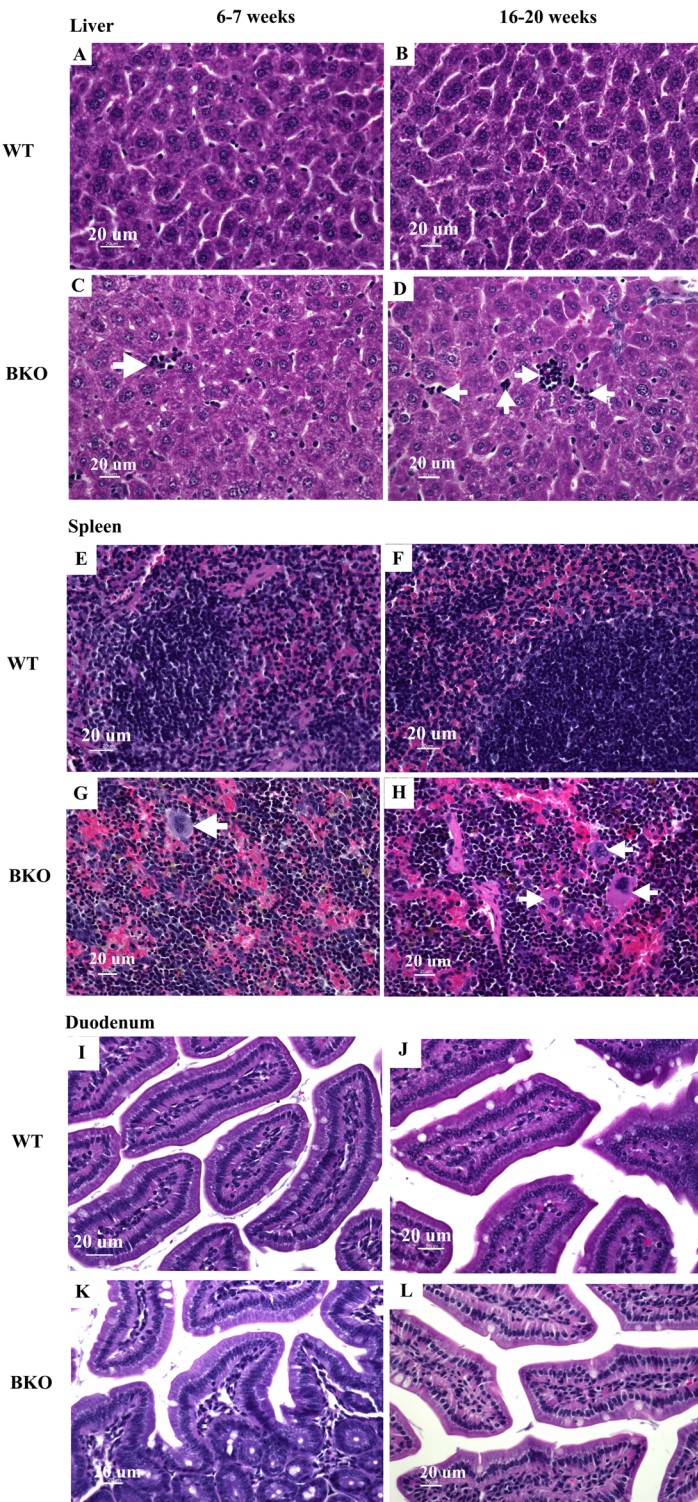

**Figure 2 Hematopoietic and mononuclear cell infiltration in the liver and spleen of BKO mice.** Hematoxylin and eosin staining of representative paraffin-embedded tissue sections from (A–D) the liver, (E–H) spleen and (I–L) duodenum of male WT and BKO mice aged 6–7 weeks and 16–20 weeks. The arrows indicate mononuclear cells and hematopoietic cells in the liver and spleen (Nikon ECLIPSE 80i light microscope; magnification ×400).

**Table 4 Mononuclear cells and hematopoietic cells score in the liver of male wild type and thalassemic mice at the age of 6–7 weeks and 16–20 weeks.**

| | WT | | BKO | | P values (2-way ANOVA) | | |
|---|---|---|---|---|---|---|---|
| | 6–7 weeks | 16–20 weeks | 6–7 weeks | 16–20 weeks | Age | Phenotype | Age × phenotype |
| Mononuclear cells and hematopoietic cells score | 0.12 ± 0.02 | 0.16 ± 0.14 | 1.86 ± 0.08[a] | 2.92 ± 0.15[b,c] | 0.0001 | <0.0001 | 0.0003 |

**Notes:**
[a] P-value < 0.05 compared with WT aged 6–7 weeks.
[b] P-value < 0.05 compared with WT aged 16–20 weeks.
[c] P-value < 0.05 compared with BKO aged 6–7 weeks.
Data are expressed as mean ± SEM ($n$ = 5/group). Statistical analysis was performed by 2-way ANOVA with Bonferroni post-hoc test.

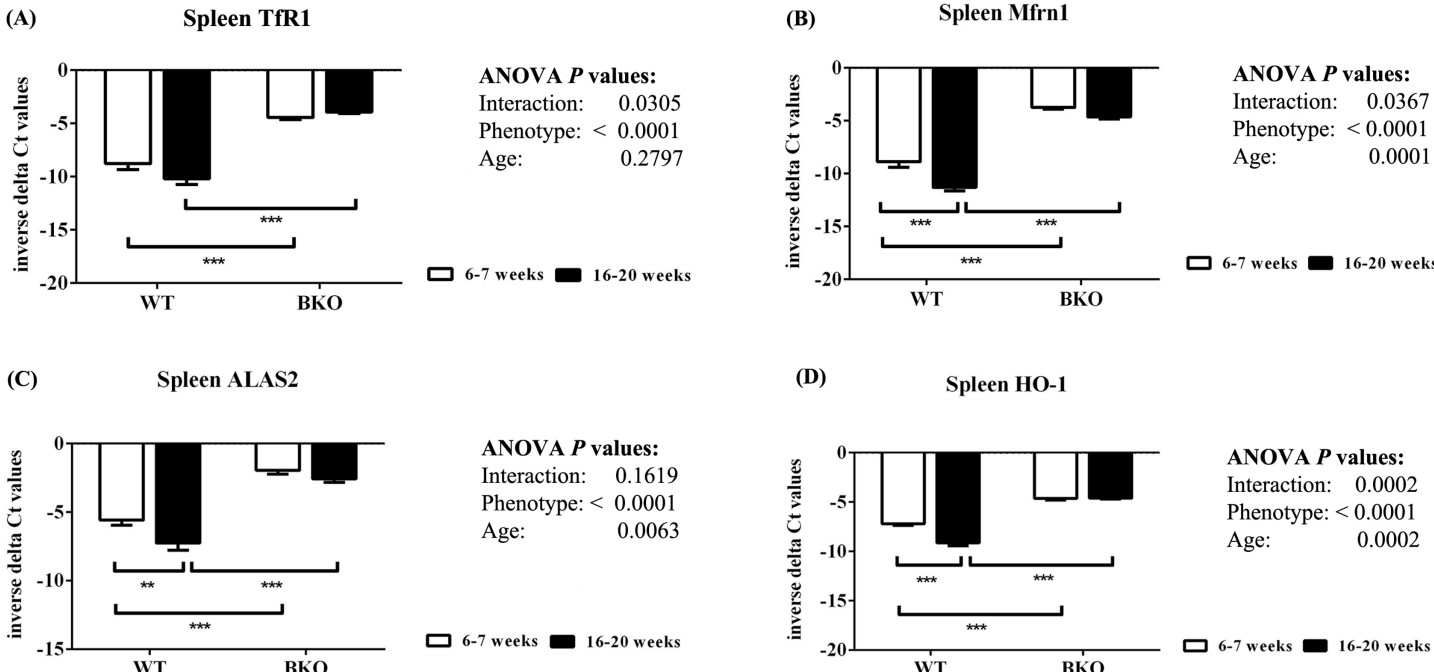

**Figure 3 The expression of genes involved in hemoglobin biosynthesis and degradation in the spleen of wild type and thalassemic mice.** Quantitative RT-PCR of (A) *Tfrc* (TfR1) (B) *Slc25a37* (Mfrn1) (C) *Alas2* (ALAS2) (D) *Hmox1* (HO-1) mRNA in the spleen of WT and BKO mice aged 6–7 weeks and 16–20 weeks. Relative mRNA expression was acquired by normalizing to *Actb* (β-actin) mRNA. Values are presented as means and SEM for minus delta Ct values ($n$ = 5 per group). Statistical analysis was performed by 2-way ANOVA with Bonferroni post-hoc test. P-values for the effects of phenotype, age and interaction are shown in the figure. Statistical significance for pairwise comparison is indicated by * symbols (**$P$ < 0.01, ***$P$ < 0.001).

and BKO mice than previously reported values (*De Franceschi et al., 2006*; *Weizer-Stern et al., 2006*; *Gardenghi et al., 2007*; *Vogiatzi et al., 2010*; *Nai et al., 2012*; *Gelderman et al., 2015*; *Kautz et al., 2015*; *Li et al., 2017*). However, comparable hematological parameters were also reported in 20-week-old WT and BKO mice obtained from the same animal facility as our study (*Wannasuphaphol et al., 2005*). Furthermore, we found that adolescent WT mice had significantly lower RBC count, hemoglobin and hematocrit along with significantly higher MCV and RDW than adult WT mice. Physiological changes in hematological parameters of young mice including lower hemoglobin, lower hematocrit, higher MCV have previously been documented (*Everds, 2007*). In agreement

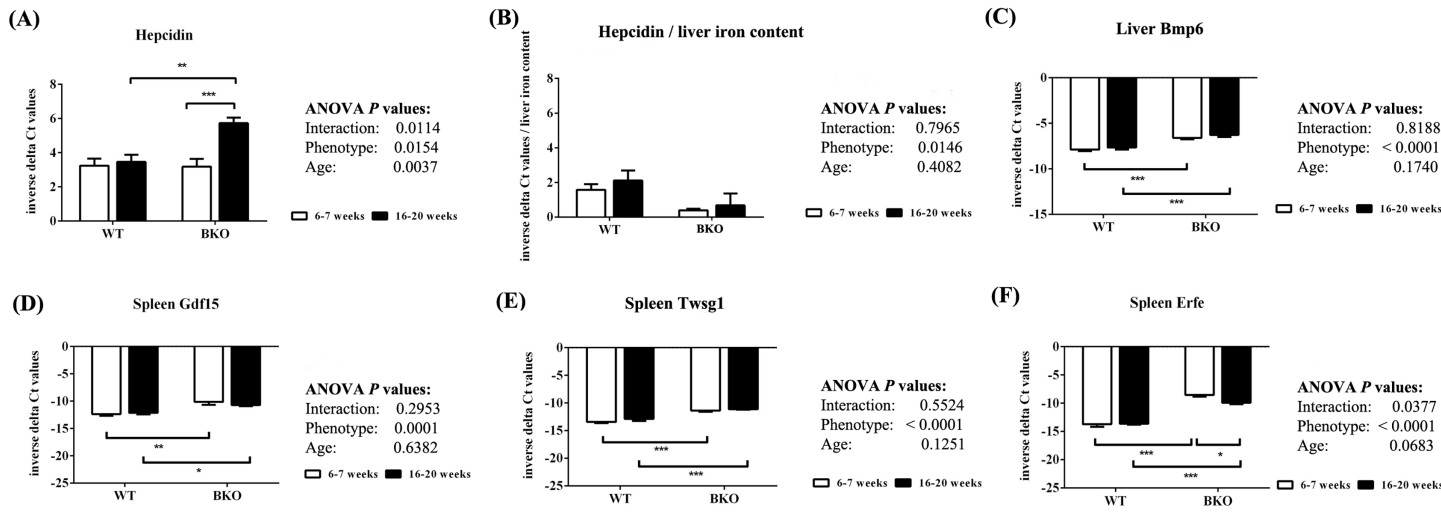

**Figure 4** **The expression of hepcidin and upstream regulators of hepcidin in wild type and thalassemic mice.** Quantitative RT-PCR of (A) *Hamp* (hepcidin) mRNA in the liver (B) Liver hepcidin mRNA relative to liver iron content (C) *Bmp6* (Bmp6) mRNA in the liver (D–F) upstream regulators of hepcidin (*Gdf15*, *Twsg1*, *Fam132b*) mRNA in the spleen of WT and BKO mice aged 6–7 weeks and 16–20 weeks. Relative mRNA expression was acquired by normalizing to *Actb* (β-actin) mRNA. Values are presented as means and SEM for minus delta Ct values ($n$ = 5 per group). Statistical analysis was performed by 2-way ANOVA with Bonferroni post-hoc test. *P*-values for the effects of phenotype, age and interaction are shown in the figure. Statistical significance for pairwise comparison is indicated by * symbols (*$P < 0.05$, **$P < 0.01$, ***$P < 0.001$).

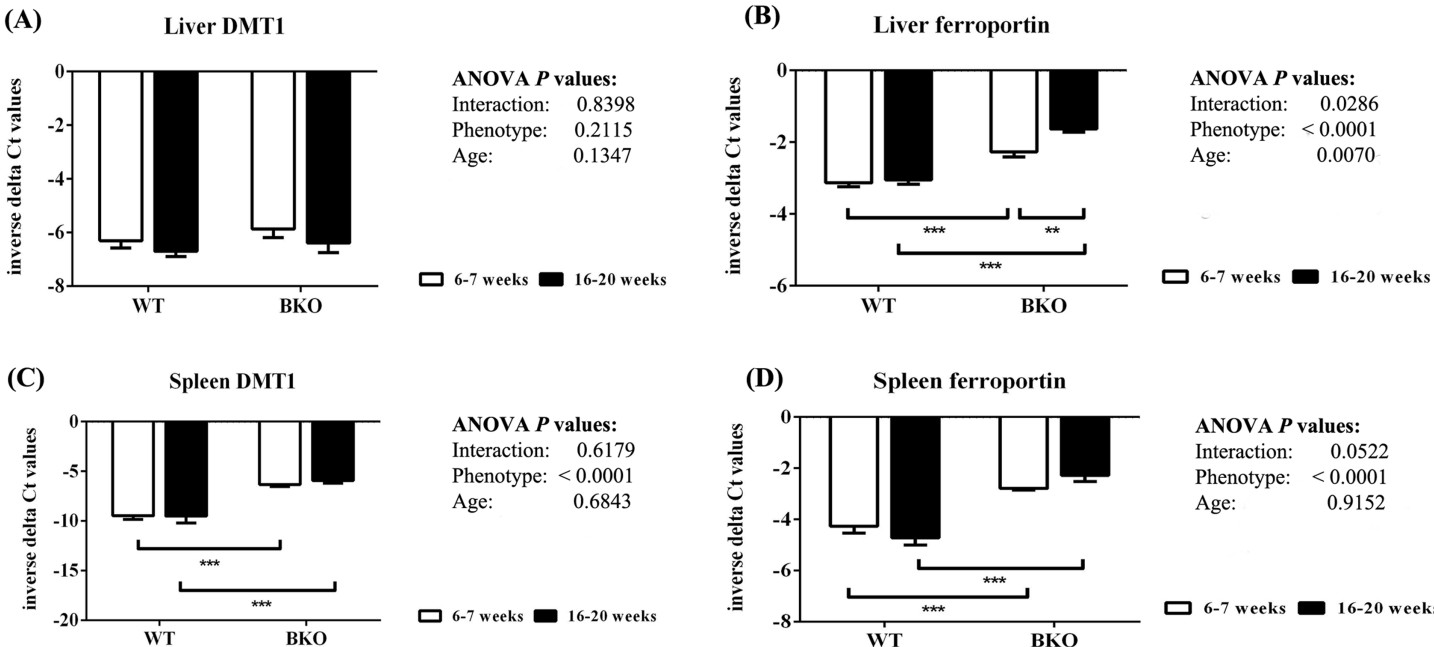

**Figure 5** **The expression of major iron transporters in the liver and spleen of wild type and thalassemic mice.** Quantitative RT-PCR of (A) Liver *Slc11a2* (DMT1) mRNA (B) Liver *Slc40a1* (ferroportin) mRNA (C) Spleen *Slc11a2* (DMT1) mRNA (D) Spleen *Slc40a1* (ferroportin) mRNA of WT and BKO mice aged 6–7 weeks and 16–20 weeks. Relative mRNA expression was acquired by normalizing to *Actb* (β-actin) mRNA. Values are presented as means and SEM for minus delta Ct values ($n$ = 5 per group). Statistical analysis was performed by 2-way ANOVA with Bonferroni post-hoc test. *P*-values for the effects of phenotype, age and interaction are shown in the figure. Statistical significance for pairwise comparison is indicated by * symbols (**$P < 0.01$, ***$P < 0.001$).

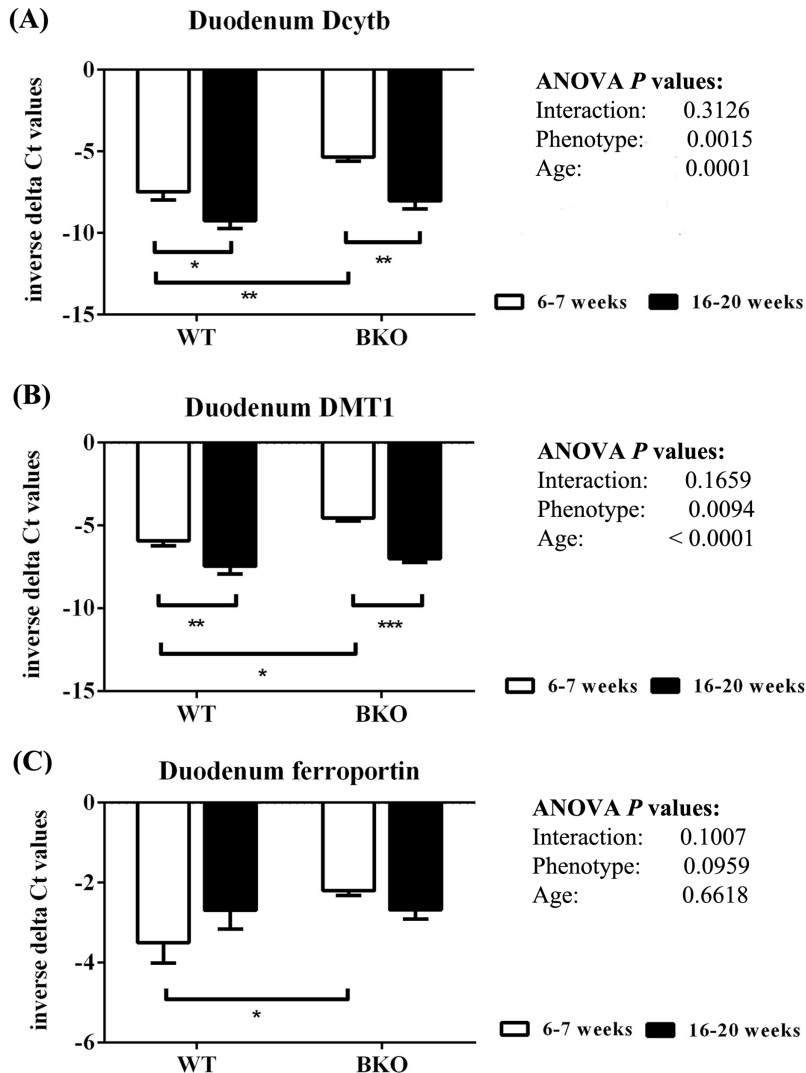

**Figure 6 The expression of major iron transport machineries in the duodenum of wild type and thalassemic mice.** Quantitative RT-PCR of (A) *Cybrd1* (Dcytb) (B) *Slc11a2* (DMT1) (C) *Slc40a1* (ferroportin) mRNA in the duodenum of WT and BKO mice aged 6–7 weeks and 16–20 weeks. Relative mRNA expression was acquired by normalizing to *Actb* (β-actin) mRNA. Values are presented as means and SEM for minus delta Ct values ($n$ = 5 per group). Statistical analysis was performed by 2-way ANOVA with Bonferroni post-hoc test. $P$-values for the effects of phenotype, age and interaction are shown in the figure. Statistical significance for pairwise comparison is indicated by * symbols (*$P < 0.05$, **$P < 0.01$, ***$P < 0.001$).

with our results on RDW values, variable red blood cells morphology has also been noted in young mice (*Bannerman, 1983*). In contrast to WT mice, the hematological abnormalities in BKO mice persisted in both age groups indicating for the persistence of thalassemic phenotype in these mice. In addition to age-associated hematological changes, increased serum iron levels during adolescence were observed in both WT and BKO mice. A similar trend has previously been reported in normal mice and rats

(*Lesbordes-Brion et al., 2006*; *Kong et al., 2015*). We speculate that serum iron levels were increased during adolescence as a result of increased iron requirement for erythropoiesis.

Several proteins are involved in the synthesis of hemoglobin in red blood cells. TfR1 is the essential iron acquisition machinery of maturing erythroid cells (*Kong, Gao & Chang, 2014*; *Gammella et al., 2017*), while Mfrn1 and ALAS2 are required for heme biosynthesis (*Hentze, Muckenthaler & Andrews, 2004*; *Amigo et al., 2011*; *Chiabrando, Mercurio & Tolosano, 2014*). The expression of Mfrn1 and ALAS2 is transcriptionally regulated by transcription factor GATA-1 during erythroid maturation (*Amigo et al., 2011*; *Tanimura et al., 2016*). In the present study, quantitative real-time PCR reveals that genes involved in heme biosynthesis and degradation were upregulated in spleen of BKO mice. Interestingly, adulthood was associated with reduced splenic mRNA expression of these genes in WT mice whereas the mRNA levels in adult BKO mice were persistently induced. In accordance, splenic mRNA levels of DMT1, which plays the central role in erythroid iron transport across endosomes (*Canonne-Hergaux et al., 2001*), were also induced in adolescent and adult BKO mice. These findings indicate that extramedullary hematopoiesis in BKO mice was induced during both age groups. In accordance, histopathological examination also showed that the extent of extramedullary hematopoiesis in BKO mice was progressively increased with increasing age. Furthermore, the presence of anemia despite enhanced extramedullary erythropoiesis in BKO mice is indicative for ineffective erythropoiesis.

In this study, both tissue non-heme iron measurement and histopathological examination revealed progressive iron accumulation in the spleen of thalassemic mice particularly in adulthood. In contrast, liver iron deposition was not significantly altered between adolescent and adult BKO mice. Notably, *Gardenghi et al. (2007)* reported a differential pace of iron accumulation in BKO mice between the spleen and hepatic Kupffer cells. In accordance with our findings, the previous work by *Kautz et al. (2015)* demonstrated that iron content in the liver of BKO mice was markedly increased during early adolescence and reached plateau during late adolescence and early adulthood. The alteration in the pattern of tissue iron deposition between adolescence and adulthood suggests that systemic iron homeostasis could differ with age.

The expression of hepcidin, the systemic iron regulatory peptide, is regulated by several factors. Under thalassemic conditions, hepcidin might be concurrently affected by systemic iron loading, anemia and ineffective erythropoiesis. It has been proposed that although erythropoiesis, iron status and inflammation all contribute to variation in hepcidin expression, erythropoietic drive appears to be the strongest contributor (*Karafin et al., 2015*). In agreement with the work by *Gardenghi et al. (2007)*, our data suggests that hepcidin expression is determined by both anemia and iron overload which have opposing effects. Furthermore, the net effect of these two factors on hepcidin expression may vary depending on age. In this study, the expression of Bmp6, a positive hepcidin regulator, and erythroid-secreted hepcidin suppressors (Erfe, Gdf15 and Twsg1) was induced in thalassemic mice. The lower hepcidin mRNA levels relative to liver non-heme iron contents in BKO mice suggest that hepcidin was influenced by erythroid regulators under thalassemic condition. On the contrary, liver hepcidin mRNA levels which

reflect the net effects of hepcidin regulators did not differ between WT and BKO mice during adolescence. Therefore, we speculate that Bmp6 and erythroid regulators both contributed on hepcidin expression in adolescent thalassemic mice. In accordance with previous studies, we observed increased hepcidin expression in thalassemic mice during adulthood (*Nai et al., 2012*; *Kumfu et al., 2016*) compared to adolescence. However, the hepcidin-to-liver iron ratio in BKO mice did not alter between the two age groups suggesting that the increase in hepcidin mRNA levels was appropriate for the increase in liver iron contents. Additionally, we observed decreased splenic Erfe mRNA expression in adult BKO mice compared to adolescent counterpart. Interestingly, Erfe has recently been demonstrated to inhibit BMP6-induced hepcidin expression (*Arezes et al., 2018*). Accordingly, the relative reduction in Erfe mRNA expression in thalassemic mice during adulthood would allow hepcidin induction by Bmp6 in response to increased liver iron accumulation. Notably, the previous study demonstrated that Erfe mRNA expression was consistently induced in the spleen of thalassemic mice during the age of 3–12 weeks (*Kautz et al., 2015*). A study to further explore whether Erfe mRNA levels in the spleen is altered in adult and aged thalassemic mice should be conducted.

We demonstrated that ferroportin mRNA expression in the spleen was induced in BKO mice and corresponded with the induction of genes involved in hemoglobin synthesis and degradation. Therefore, it is possible that increased erythropoiesis and the subsequent degradation of erythroid cells are responsible for the increased splenic ferroportin expression in adult BKO mice. In agreement, erythrophagocytosis has been shown to induce ferroportin and HO-1 transcription in vitro (*Knutson et al., 2003*; *Delaby et al., 2008*). Furthermore, the expression of ferroportin in iron-recycling macrophages is upregulated by both heme and iron acquired from the degradation of hemoglobin in senescent red blood cells. Heme has been shown to transcriptionally induce ferroportin whereas the regulation of ferroportin expression by iron occurs at the translational level through IRP-IRE interaction (*Marro et al., 2010*; *Drakesmith, Nemeth & Ganz, 2015*). In the liver, ferroportin mRNA expression was induced in adolescent and adult thalassemic mice, possibly, as a result of liver iron accumulation, which was present in both periods of lifespan. In line with our findings, iron dextran administration has been shown to induce liver ferroportin mRNA expression in C3HeB/FeJ mice (*Liu et al., 2005*). In contrast to ferroportin, we found that DMT1 mRNA expression in the liver was not affected by thalassemia or age. In thalassemia, the increase in iron levels in the circulation exceeds the binding capacity of apotransferrin resulting in the presence of non-transferrin bound iron (NTBI). Notably, a previous study reported that DMT1 does not play an essential role in the uptake of NTBI in the liver (*Wang & Knutson, 2013*). Further studies should be performed to delineate the roles of other NTBI transporters such as Zip14 in the pathophysiology of liver iron overload under thalassemic condition and to determine whether the expression of these transporters differs with age.

Intestinal iron hyperabsorption as a result of hepcidin downregulation has previously been reported in thalassemic mice (*Gardenghi et al., 2007*). In agreement, we demonstrated significant effects of thalassemia on the mRNA expression of apical iron transport

machineries, Dcytb and DMT1, in the duodenum. Interestingly, reduced Dcytb and DMT1 mRNA expression was observed in the duodenum of adult WT and BKO mice compared to adolescent counterparts. These changes in duodenal iron transporter expression also coincided with the reduction in serum iron levels of both WT and BKO mice during adulthood. It has previously been shown that hepcidin expression inversely correlates with duodenal iron transporters mRNA expression and iron absorption in rats (*Frazer et al., 2002*, *2004*; *Millard et al., 2004*). However, our study observed Dcytb and DMT1 downregulation in adult WT mice compared to adolescent WT mice although hepcidin expression was unaltered between these two age groups. Therefore, we speculate that hepcidin did not directly involve in the altered Dcytb and DMT1 expression between adolescence and adulthood. A previous study in rat reported that the expression of Dcytb, DMT1 and ferroportin was significantly affected by age (*Kong et al., 2015*). Moreover, hypoxia-inducible factor 2 alpha (HIF-2α) has been shown to transcriptionally induce Dcytb and DMT1 in response to iron deficiency (*Mastrogiannaki et al., 2009*; *Shah et al., 2009*). In the present study, we not only found anemia in BKO mice but also reduced levels of RBC count, hemoglobin and hematocrit in adolescent WT mice compared to adult WT mice. It is, therefore, possible that anemia-associated hypoxia was responsible for the upregulation of intestinal iron transporters observed in our adolescent mice. The mechanisms of age-associated changes in duodenal iron transporters including the possible roles of HIF-2α remain to be elucidated.

In summary, our study found that several hematological and iron homeostatic parameters were different between adolescence and adulthood. As the mice progressed from adolescence to adulthood, duodenal iron transporter mRNA expression and serum iron levels of both WT and BKO mice were decreased. Reduced splenic mRNA expression of genes involved in hemoglobin metabolism as well as alteration in erythrocyte parameters were found only in WT mice. In BKO mice, erythrocyte abnormalities along with induction of genes involved in hemoglobin metabolism in the spleen were persistent in both adolescence and adulthood. Interestingly, adulthood was associated with increased liver hepcidin and ferroportin mRNA expression along with splenic Erfe mRNA suppression only in BKO mice. The limitations of the present study include the lack of findings on serum hepcidin and Erfe levels as well as gene expression in the bone marrow. In addition, the expression of iron transporters at the protein level should be performed in future studies.

## CONCLUSIONS

Our study shows that iron homeostasis in a mouse model of thalassemia intermedia differs between adolescence and adulthood. During adolescence, increased mRNA expression of duodenal iron transporters is associated with liver iron accumulation. In adult BKO mice, the induction of splenic Erfe and duodenal iron transporters is diminished compared to adolescent BKO mice. Furthermore, extramedullary erythropoiesis is enhanced and the spleen becomes preferential site of tissue iron loading. The present study demonstrates that iron homeostasis in thalassemia intermedia might be altered with the age per se as well as disease progression. This study, thus, underscores the significance of age on the

expression of genes involved in iron metabolism as well as the pathophysiology of iron loading in thalassemia intermedia. Therefore, the age of thalassemic mice should be considered in the study of iron homeostasis under thalassemic condition.

## ACKNOWLEDGEMENTS

We would like to thank Thalassemia Research Center, Institute of Molecular Biosciences, Mahidol University for supplying thalassemic mice.

### Funding

This work was supported by Office of the Higher Education Commission; The Thailand Research Fund; Mahidol University; Siriraj Research Fund, Faculty of Medicine Siriraj Hospital, Mahidol University; National Research Council of Thailand (NRCT). Chanita Sanyear was supported by the Royal Golden Jubilee (RGJ) scholarship from The Thailand Research Fund (PHD/0052/2556) and National Research University (NRU) scholarship, Thailand. The funders had no role in study design, data collection and analysis, decision to publish, or preparation of the manuscript.

### Grant Disclosures

The following grant information was disclosed by the authors:
Office of the Higher Education Commission.
The Thailand Research Fund.
Mahidol University.
Siriraj Research Fund, Faculty of Medicine Siriraj Hospital, Mahidol University.
National Research Council of Thailand (NRCT).
The Thailand Research Fund (the Royal Golden Jubilee scholarship): PHD/0052/2556.
National Research University (NRU) scholarship, Thailand.

### Competing Interests

The authors declare that they have no competing interests.

### Author Contributions

- Chanita Sanyear conceived and designed the experiments, performed the experiments, analyzed the data, prepared figures and/or tables, authored or reviewed drafts of the paper, and approved the final draft.
- Punnee Butthep conceived and designed the experiments, prepared figures and/or tables, and approved the final draft.
- Wiraya Eamsaard performed the experiments, prepared figures and/or tables, and approved the final draft.
- Suthat Fucharoen conceived and designed the experiments, prepared figures and/or tables, and approved the final draft.
- Saovaros Svasti conceived and designed the experiments, prepared figures and/or tables, and approved the final draft.

- Patarabutr Masaratana conceived and designed the experiments, performed the experiments, analyzed the data, prepared figures and/or tables, authored or reviewed drafts of the paper, and approved the final draft.

## Animal Ethics

The following information was supplied relating to ethical approvals (i.e., approving body and any reference numbers):

Institute of Molecular Biosciences Animal Care and Use Committee, Mahidol University, Thailand approved the study (COA.NO.MB-ACUC 2016/003).

## Data Availability

The raw data for the tables and figures is available in the Supplemental Files.

## Supplemental Information

Supplemental information for this article can be found online at http://dx.doi.org/10.7717/peerj.8802#supplemental-information.

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
