# Peer review of "Iron homeostasis in a mouse model of thalassemia intermedia is altered between adolescence and adulthood"

_PeerJ, doi:10.7717/peerj.8802_

## Round 0.1 · original submission · Major Revisions

Your manuscript is interesting in the field and readers. I hope you can easily improve the quality of your original manuscript. Looking forward to seeing your revised version

Reviewer 1 ·

Basic reporting

This work investigates age-dependent differences in iron metabolism in the Th3/+ mouse model of thalassemia. The authors report changes in serum and tissue iron parameters, as well as in expression of iron genes between young and adult mice. The findings are interesting and adequately discussed.

Experimental design

My major concern is that it is not clear which of the reported changes are specific to thalassemia and which (if any) are related to age-dependent variations that also occur in wild type animals. Thus, it will be important to present hematological data and serum/tissue iron data in wild type mice. I would also suggest to report the qPCR data of wild type and Th3/+ mice side by side, instead of normalizing the Th3/+ data relative to wild type. The way the data are presented does not allow comparison of expression levels in wild type mice of different age groups. Hepcidin mRNA data could be normalized to liver iron for clarity. Finally, the data would be strengthened by showing serum hepcidin levels, as well as expression of proteins encoded by the shown mRNAs (at least for the ones where antibodies are commercially available).

Validity of the findings

The findings are generally appropriately discussed. However, the discussion should be modified if age-dependent differences in wild type mice are identified. In the Discussion (lanes 338-341) the authors argue that the age-dependent differences in ferroportin mRNA are linked to hepcidin. This is completely wrong! Hepcidin controls ferroportin protein, not mRNA expression.

Additional comments

There are several statements in the Introduction which are inaccurate. For instance:
a) lanes 76-78: If there is indeed any evidence that DMT1 mediates NTBI uptake in hemochromatosis or thalassemia, references should be provided. To the best of my knowledge, experiments with liver-specific Dmt1-/- mice have excluded any role of this transporter in iron overload (PMID 23508576).
b) lanes 86-87: The rate-limiting step for heme biosynthesis in non-erythroid cells is the generation of aminolevulinic acid, catalyzed by ALAS1. By contrast, the rate-limiting step for heme biosynthesis in erythroid cells is iron uptake (PMID 10522552).
c) lanes 90-91: “IRP” stands for “iron regulatory protein”
d) lane 92: ALAS2 is not an iron transporter

Reviewer 2 ·

Basic reporting

I. BASIC REPORTING:
Clear, unambiguous, professional English language used throughout:
No

1. Specific comments
The standard of English language used is generally satisfactory in most parts of the manuscript, but there are parts where language editing is required for grammatical correctness and better clarity.

Intro & background to show context
2. Specific comments on the introduction:
A lot of the introduction section is a very generalised review of iron homeostasis, which is unnecessarily long. A more concise account would suffice.

3. Specific comments on the background to show context
The rationale and context that prompted the work done is not clear. This should be clearly explained.

Literature well referenced & relevant.
No.

4. Specific comments
Page 7, lines 49 -56:
The first paragraph of this section gives an overview of thalassemia, stating facts that are available in any standard textbook on the subject. The authors have provided, as a reference for these statements, an article published in 2013 in a journal that is not of high quality. More appropriate and better quality references should be provided in support of these statements.

5. Page 7, lines 57 -58:
“Iron, one of essential trace elements, is involved in several biological processes such as oxidative phosphorylation and hemoglobin synthesis (Hentze et al., 2004).”
The fact, stated in the above sentence, is almost common knowledge for anybody in the field of biology. This statement, therefore, needs to be supported by a more appropriate reference than a review article published in 2004, as done by the authors.

6. Page 7, lines 66-78: These sentences outline basic facts about iron absorption. No references have been provided for any of the statements made. Appropriate references need to be provided for the whole of this paragraph.

7. Page 8, lines 90-96: References need to be provided for these statements.

Experimental design

Research question well defined, relevant & meaningful. It is stated how the research fills an identified knowledge gap.
No.

8. Specific comments
The rationale and context that prompted the work done is not clear. What was the research question the authors hoped to address? What is the relevance of such information, in the context of thalassemia?

9. Age-related changes in events involved in iron homeostasis and in thalassemic mice have been reported before (De Franceschi et al., 2006; Chen et al, 2009; Gardenghi et al, 2007). These studies have been cited by the authors of this manuscript. What is different about what the authors of the present manuscript have studied or found from their work? The authors should clearly state what lacunae their work addressed and why it was important to address these.

Rigorous investigation performed to a high technical standard.
10. No. The authors have carried out only gene expression studies done, when protein expression studies would have been more relevant and appropriate.

Methods described with sufficient detail & information to replicate.
No.

11. Specific comments
Methods:
The authors should provide references to support their rationale to use mice aged
6-7 weeks old, 8-12 weeks old, and 16-20 weeks in their study. Mice in the first 2 groups can actually be very close in age. For example, a 7-week old mice would be expected to be very similar to an 8-week old mouse, but they fall into 2 different groups as per the methodology used by the authors. Perhaps a wider difference in the ages may have been a better strategy to use. The authors should substantiate their rationale for choosing these ages to study, and the importance and relevance of parameters of interest in adolescent, young adult and adult mice.

12. What was the source and iron content of the chow used to feed the mice?

13. Were age and gender-matched WT mice also studied for purposes of comparison? There is no mention of such mice in the methodology section. Such comparisons would be necessary in the context of this study.

14. How many mice were used in each of the age groups studied and for each parameter? There is no information provided about this.

15. Why was ANOVA chosen for analysis? The numbers are small and the data could hardly be normally distributed. Non-parametric tests are more appropriate in such situations.

Validity of the findings

16. Table 2: How do the authors explain their observation the liver iron content was lower in the older mice than the youngest ones, with levels at 8 to 12 weeks being lowest?

17. The legend for the table should state n=? and what statistical tests were used for the analyses.

18. Data for all the parameters studied should have been shown for age-matched WT mice, to make meaningful comparisons.

19. Page 11, lines 187-189: It is stated “Hematological parameters of BKO mice demonstrated thalassemia phenotype including low hemoglobin, hematocrit, MCV and RBC count along with high RDW (Table 2).”
What is the comparison group the authors have used to state that the parameters, listed in the above sentence, above were low? Comparison with age and gender-matched WT mice would be necessary to make such a statement. Such data has not been shown. The authors should show these data.

20. Page 11, lines 189-190: It is stated “No significant difference in RBC count, hemoglobin and hematocrit was found.” Please state clearly which groups are being compared when making this statement.

21. Table 3: Please explain how it is that the scores shown in this table are just these single numbers.
The reference, quoted for the methodology used to get the scores shown in Table 3, gives scores based on percentage of hepatocytes that contained iron. How many fields were studied for each section? How were these fields chosen? What numbers of hepatocytes were counted in each field? What was the mean score obtained? What was the SD? How many mice were used for these studies?
The authors should provide all this information.

22. Page 11, lines 198-199: It is stated “….Additionally, a slightly positive iron staining was observed in brush border of duodenal enterocytes…..”
What does “a slightly positive iron staining” mean? This is an ambiguous statement. Please clarify.

23. Page 11, lines 200-201: It is stated “..An increase in iron deposition score was found in the spleen of the 16-to-20-week old BKO group …...”.
To what group is the data being compared to make this statement?

24. Page 12, lines 206-207: It is stated “Microscopic examination of liver and spleen tissue samples revealed the presence of extramedullary hematopoiesis in BKO mice (Fig. 2)….”.
Please describe what features in these sections indicate “the presence of extramedullary hematopoiesis”. This figure also shows H and E sections of the duodenum. Do the authors mean to say that extramedullary hematopoiesis is seen in the duodenum also?

25. Legends for figures 3 and 4 state that n= 5 for each group. With such small numbers, use of ANOVA is not appropriate. Non-parametric tests should have been employed for analyses. The authors should re-analyse their data to see what their results really show.

26. Why did the authors not study erythroid regulators of hepcidin in the bone marrow, in addition to the extra-medullary sites, in their experimental mice? Erythroid factors produced during ineffective erythropoiesis in the bone marrow may have profoundly affected hepcidin expression in the thalassemia mouse model they studied. How have they ruled this out?

27. Fig 4: The authors show increased hepcidin expression in the liver in the 16-20 weeks-old BKO mice. However, there is no increase in BMP6 expression in the liver, or significant decreases seen in the erythroid factors in the liver, which are the expected biological changes associated with increased hepcidin expression. How do the authors explain their findings?

28. There is some decrease in expression of these erythroid factors in the spleen. If the authors mean to link these changes to the hepcidin expression in the liver, they needed to have measured blood levels of these regulators and also their expression levels in the bone marrow. Such data are necessary to suggest such an association.

29. What were the serum concentrations of hepcidin in these mice?

30. How do the authors interpret their findings in Fig 4?

31. Figure 5: There are 2 bar charts each in Fig 5A and 5B. One cannot give 2 figures the same number. These need to be re-named appropriately for greater clarity. The text that describes these results also lack clarity.

32. Page 13, lines 237-238:
It is stated “Liver ferroportin mRNA levels of BKO mice were 2-3 times higher than age-matched WT mice (Fig. 5A).”
If one looks at the data shown for ferroportin in the figure, the above sentence is not quite an accurate statement of the results shown. The authors have made an overall generalized statement. This is the results section, not a summary of their results. In view of this, it is important that they state their results more accurately.

33. Page 13, lines 238-239:
The authors go on to state “It is noteworthy that ferroportin mRNA expression was marginally induced with increasing age although the difference was not statistically significant.”
Once again, this is not a factually correct statement of the data shown for ferroportin in the figure. Why do the authors think this is “noteworthy”?

34. Page 13, lines 243-245:
“Furthermore, splenic ferroportin mRNA levels in BKO mice were significantly increased in the 16-to-20-week old group than the younger age groups.”
How do the authors interpret this observation, in the view of the increased hepcidin expression they report in the liver? In this context, it would be relevant to know what the serum levels of hepcidin are in these mice.

35. Page 13, lines 247-248:
“…….shown in Fig. 6, duodenal DMT1 and ferroportin mRNA expression was significantly suppressed in 16-to-20-week old BKO mice compared to the other BKO groups”
How do the authors interpret these observations?

36. The effect of hepcidin on ferroportin has been shown to be at the post-translational level (Drakesmith et al, Cell Metab 2015 [review]) and DMT1 (Brasse-Lagnel C et al, Gastroenterology. 2011; Yamaji et al, Blood 2004). Studying protein expression of these proteins would have been the more meaningful analyses to have carried out in the context of this study. Simply making observations of mRNA expression, without accompanying data on protein expression or functionality, is not meaningful.

Additional comments

DISCUSSION
Overall, the discussion section of this manuscript is not well-written.

27. Page 13, lines 253-256:
It is stated “Several factors including age, gender and strain have been shown to influence iron parameters as well as the expression of hepcidin and other iron regulatory genes (Courselaud et al., 2004; Krijt et al., 2004; Weizer-Stern et al.2006; McLachlan et al., 2017).”
All the above references show the effect of gender and strain on hepcidin and iron-related parameters. They have not studied the effect of age on these parameters. Hence, it is factually incorrect to cite the above references in the context of studying the effect of “age”.

28. The present study appears to be similar in many ways to part of the work done by Gardenghi et al (Blood 2007), where they studied mice (similar to those in the present study) across a much wider range of ages. What new information does the present study provide across a very narrow range of ages and why did the authors think this was important to study? This is unclear.

29. Page 13, lines 261 to 263:
“….Moreover, iron status and iron homeostasis might be altered in adolescents as a result of increased body iron requirement for growth and development.”
Please provide references in support of the above statement.

30. Page 14, lines 263 to 264:
“However, it has not been clearly elucidated whether iron homeostasis during adolescence differ from adulthood particularly under thalassemic condition….”
The above statement is not factually correct. Gardenghi et al (Blood 2007) have studied the same model as in the present study, where they studied such mice at 2, 5, and 12 months of age. The first 2 time periods they studied (more than 12 years ago) have similarities to the time periods in the present study (6-7 weeks, 8 to 12 and 16 to 20 weeks). Hence, the above statement is not justified.

31. The reference (Laviola et al., 2003) cited by the authors in support of the ages of mice they have used states:
“…….In rodents, the term ‘adolescence’ covers the whole postnatal period ranging from weaning (PND 21) to adulthood (PND 60), that is to be considered no longer infancy, but not yet adulthood. This is indeed a period of behavioral transition, which has been adopted by several authors as a suitable animal model in order to study the psychobiological characteristics of human adolescence……an adolescent rodent has been classified by the use of three age-intervals, namely early adolescence (prepubescent or juvenile, PND 21-to34), middle adolescence (periadolescent, PND 34-to-46), and late adolescence (young adult, PND 46-to-59).…..”

32. According to the above statements, mice in 2 of the 3 groups studied (6 to 7 weeks and 8 to 12 weeks) fall into the adolescent category. What was the purpose of these choices of ages for the mice in this study?
The relevance of using these specific ages is not quite clear. What is the relevance to be able to designate mice as being adolescent or early adults or adults and study the parameters of interest at these time points?

33. Page 14, lines 275- 279: It is stated “In the present study, hematological results revealed relatively lower red blood cell count and hemoglobin levels in BKO mice than previously reported values (De Franceshi et al., 2006; Weizer-Stern et al., 2006; Gardenghi et al., 2007; Vogiatzi et al., 2010; Nai et al., 2012; Gelderman et al., 2015; Kautz et al., 2015; Li et al., 2017). Similar trends were also observed in WT mice (data not shown).”
Data on WT mice need to be shown.

34. Page 14, lines 279- 282: It is stated “….It is noteworthy that the study conducted in 20-week old WT and BKO mice obtained from the same animal facility as our study also reported comparable hematological parameters to our finding (Wannasuphaphol et al., 2005).”
The authors should explain why they consider this observation “noteworthy”.

35. Page 14, lines 282- 283: It is stated “….Although significant differences in MCV, MCHC and RDW across the studied age ranges were found, similar findings were also observed in WT mice and appeared to be transient.”
It is not acceptable to simply make such statements without showing data for WT mice. Readers need to see the data to decide for themselves the credibility of such statements.

36. Page 14, lines 284- 286: It is stated “…In this study, both tissue non-heme iron measurement and histopathological examination revealed progressive iron accumulation in the spleen of thalassemic mice particularly during adulthood whereas liver iron deposition appeared to be generally steady….”
What does the statement “…liver iron deposition appeared to be generally steady…” mean? Please define “generally steady”. What is the data that this statement is based on?

37. Page 14, lines 286- 288: “….Furthermore, a pronounced increase in spleen iron content was observed in BKO mice during late adolescence and early adulthood…..”

And

Page 15, lines 293- 294: “….the pattern of tissue iron deposition between adolescence and adulthood suggests that systemic iron homeostasis could differ with age….”
What do the authors think is the relevance or importance of such changes at these specified time periods?

38. Page 15, lines 309 - 312: “….Our data is in agreement with the work by Gardenghi et al. (Gardenghi et al., 2007) which proposed that the expression of hepcidin may be determined by the relative ratio between anemia and iron overload which may vary depending on age.”
What is mean by the terms “relative ratio between anemia and iron overload”?
It is also not clear how the authors of the present study make the claim they have in the above sentence; please explain.

39. What were the changes in relevant proteins involved in erythropoiesis in the bone marrow, when this site is a major site of the erythroid factors that suppress hepcidin? Why did the authors study only extra-medullary erythropoiesis and not the bone marrow also?

40. Page 16, lines 333 - 335: “….On the other hand, hepcidin expression is induced in
adulthood leading to relatively normalized intestinal iron absorption and relatively stable liver iron accumulation.”
How do the authors explain the induction of hepcidin in adulthood? What is the basis of their saying that this supposed induction led to “….relatively normalized intestinal iron absorption and relatively stable liver iron accumulation…”? If hepcidin was induced, it should have resulted in decreased protein expression of DMT1 and ferroportin in the duodenum. Without data on serum hepcidin concentrations and protein expression of duodenal DMT1 and ferroportin, how do the authors make these claims?

41. Page 16, lines 335 - 337: “….Additionally, the increased erythropoiesis in adult BKO mice could lead to splenic iron acquisition from senescent red blood cells and, thus, resulting in the shift of predominant site of tissue iron accumulation from the liver to the spleen.”
What do the authors base the above statement on?

42. Page 16, lines 338 - 339: “…It is also noteworthy that ferroportin mRNA expression was affected by age in tissue-specific manner….”
What is the basis of the authors making such a statement?

43. Page 16, lines 339 - 341: “…The reciprocal expression between hepcidin and duodenal ferroportin suggests that the expression of ferroportin in the duodenum was under the regulation of hepcidin at least during 6-20 weeks of age….”
Ferroportin is well known to be regulated by hepcidin at the post-translational level (see Drakesmith H, Nemeth E, Ganz T. Ironing out Ferroportin.Cell Metab. 2015 Nov 3;22(5):777-87 for an excellent review). The authors have studied mRNA expression of ferroportin and attempt to explain their observations of changes in mRNA levels in the duodenum as effects of hepcidin. This is not convincing. Data on protein expression or functionality is required to support such a statement.

44. Page 16, lines 341 - 343: “…. ferroportin mRNA expression in the spleen was markedly induced in adulthood and corresponded with the expression pattern of genes involved in hemoglobin synthesis and degradation…..”
This observation can be well-explained by what is known about the transcriptional regulation of ferroportin. However, the authors have made no attempt to do so. They should refer to the review mentioned above (Drakesmith et al, 2015) to enable them to discuss their findings.

45. Page 16, lines 345 - 346: “…. it is likely that increased erythropoiesis is responsible for the increased splenic ferroportin expression in adult BKO mice…”
This is a statement that conveys a superficial understanding of what is known about these phenomena. The authors should discuss these findings (and possible mechanisms) better.

46. Page 16, lines 346 - 348: “…. In the liver, ferroportin mRNA expression was
generally induced in adolescence and adult, possibly, as a result of liver iron accumulation which were steady during this period of lifespan.”
What is the basis for stating that liver iron accumulation was “steady” in adolescent and adult mice? What does “steady” mean in this sentence?

47. On page 12, lines 207-210, it is stated “As shown in Table 4, an increasing age was associated with increased number of liver mononuclear cell infiltration and hematopoietic cells suggesting that the extent of extramedullary hematopoiesis progressively increased at least during the studied age range.”
The relevance/significance of the above observations have not been discussed. The authors should do so.



48. Increased Tfrc mRNA expression, which is one of the observations in the spleen, is an indication of iron deficiency in the cells. How do the authors interpret this observation? They have also not discussed its relevance/significance.

49. Page 16, lines 352 - 354:
“In conclusion, our study demonstrated the differences between iron homeostasis as well as the expression of genes involved in iron transport and metabolism in a mouse model of thalassemia intermedia between adolescence and adulthood.”
This is a poorly constructed sentence and need to be re-phrased for clarity.

50. Page 16, lines 354 - 356:
“During adolescence, a markedly induced Erfe expression results in hepcidin suppression, intestinal iron hyperabsorption and liver iron accumulation.”
A statement of conclusion as above is not warranted in the absence of data on ERFE in the bone marrow and serum levels of ERFE and hepcidin. They have also not shown data to support their conclusion of “intestinal iron hyperabsorption”.

51. Page 17, lines 354 - 356:
“In adulthood, extramedullary erythropoiesis is enhanced and Erfe induction is relatively diminished leading to BMP6-mediated hepcidin upregulation.”
The above statement is not warranted based on the data shown in this manuscript. No induction of BMP6 has been shown, to indicate that the upregulation of hepcidin in the liver is “BMP6-mediated”, as claimed.

52. Page 17, lines 357:
“Subsequently, intestinal iron absorption is suppressed…”.
No data on measurement of intestinal iron absorption are shown. Hence, the above statement is not warranted based on the data shown in this manuscript.

53. Page 17, lines 358 - 361:
“The present study underscores the significance of age on the pathophysiology of iron loading in thalassemia intermedia. Therefore, the age of thalassemic mice should be considered in the study of iron homeostasis under thalassemic condition.”
The authors make statements that are far too sweeping. They have also not shown anything new that was not shown in the publication by Gardenghi et al, 2007, which is a publication that they themselves have quoted.

54. The authors have provided no information on ethical approval having been obtained for the work done on the mice in the study.

55. The abstract of the manuscript:
a. The language in the manuscript needs improvement for grammatical correctness.

b. “….by iron regulatory hormone, hepcidin” to be replaced by “by iron- regulatory hormone, hepcidin…”

c. “Notably, effects of age on iron homeostasis in this mouse model particularly during adolescence have not been explored”.
The authors should mention why it would be necessary to explore this.


“Results.
d. Significant differences in the expression of hepcidin and several iron-related molecules across age groups were found.”
This is a very non-specific statement that gives little useful information. The authors should briefly state their findings.

e. “In adolescent BKO mice, ineffective erythropoiesis results in inappropriately low hepcidin expression and upregulation of duodenal iron transporters.”
What was the comparison group? This should be mentioned.

f. “On the other hand,increased expression of hepcidin and several genes involved in hemoglobin metabolism along with normalized expression of duodenal iron transporters was observed in adult BKO mice”
What was the comparison group? This should be mentioned.

Reviewer 3 ·

Basic reporting

The use of English is appropriate.
Some references are missing and these have been pointed out in comments to author.
Article structure, figures and tables are appropriate.
Raw data has been given in the supplementary material.
Results are relevant to hypothesis

Experimental design

The manuscript is within the scope of the journal
The research question is well-defined and the results fill a gap in knowledge
The experiments carried out are elegant and well designed.
Methods have been described in sufficient detail. Some comments for improvement have been given in the comments to author.

Validity of the findings

Findings are valid and robust.
The conclusion section is speculative and not entirely supported by the data. Some suggestions for improvement are given in comments to author below.

Additional comments

Abstract:
Methods section of abstract: It is not clear whether comparisons were made between BKO mice and wild-type mice of the same age, or between BKO mice at different ages.
Mention the age of mice in brackets for adolescent, young adult and adult mice.
Results section of abstract: Data are reported only for adolescent and adult mice. There is no mention of results in the young adult group.

Introduction
If Gardenghi et al 2007 have already shown that age influences iron regulatory parameters in BKO mice, then what was the rationale for doing this study? This must be clarified.
Line 50: “In thalassemia…” Change to “In beta thalassemia…”
Lines 66-78: Insert references for the statements made.
Lines 90-97: Insert references for the statements made.
Lines 114-116: May be relevant to provide some more details regarding the phenotype of the Hbbth3/+ mice. For example, what is the extent of anemia in these mice? what are the tissues that are iron overload? Is the spleen iron loaded in these mice? Is there extramedullary erythropoiesis in this model? Is there growth retardation compared to wild-type mice?
Briefly describe what is known about the effect of age on iron metabolism in wild-type mice fed a normal chow diet.
The Introduction section can be shortened by deleting some of the irrelevant portions.

Materials and methods:
Mention C57Bl/6 J or C35Bl.6 N
What is the iron content of the rodent chow that was provided to the mice?
Authors are encouraged to submit the ARRIVE checklist as a supplementary file

Results:
Tables 1 and 2: Is there data available for age-matched wild-type mice?
Table 1 shows a 50% reduction in liver iron in young adult mice. However, this reduction is not reflected in the histopathological scoring in Table 1. What is a possible reason for this?
Figure 3: From the figure legend, it is not clear how the authors calculate the fold change compared to WT mice. However, from the raw data provided in supplementary material, it is apparent that the average obtained for WT mice at each age was used for normalization. This information must be provided in the legend.
Since there are likely to be age-related changes in wild-type mice as well, it may be better to show the data for wild-type and BKO mice at each age (rather than show fold change compared to wild-type). The y-axis can show delta Ct values instead of fold changes. This will yield more information. If the authors do this, then 2-way ANOVA will be the correct statistical test.
Figure 4: Same comment as for Fig 3 regarding showing the data for WT mice. Line 221: The authors state the hepcidin mRNA was marginally lower than the corresponding WT group. This is difficult to appreciate in Fig 4 because the data for WT is not shown. It is also not clear if the decrease was statistically significant even if it is “marginal”.
Line 221-222: It is possible to show that hepcidin expression was inappropriately low for liver iron content in BKO mice by showing data for the ratio of hepcidin mRNA to liver iron content. This will also clarify if in adult mice the hepcidin expression (which was increased) was appropriate for liver iron content.
Figure 4: It would be interesting to know the actual mRNA expression levels of GDF-15, TWSG-1 and ERFE in the liver and spleen in WT mice (since the expression of these “erythroid regulators” are usually looked for in the bone marrow). This would provide baseline information in order to interpret the increase seen in BKO mice.

Discussion
Lines 265 to 271 which gives the rationale for selecting the ages of mice for this study should probably be given at the end of the introduction section or in methodology. This will make better sense as the mice are described here and an obvious question would be why these particular ages were chosen.
It would be interesting to see if Zip14 mRNA levels in the liver changes with age in WT and BKO mice
Lack of data on protein levels must be mentioned as a limitation of this paper.
Lines 301 and 306: Contrary to what the authors have stated, I would argue that the hepcidin levels are not explained by ERFE in the spleen since ERFE in spleen is decreased both in young adults and adults while hepcidin is increased only in adults. Therefore, the increase in hepcidin expression seen only in adult BKO mice is not explained by changes in ERFE in the spleen alone. Do the changes in liver hepcidin expression reflect the changes in expression of the erythroid regulators in the bone marrow? Lack of data on the expression of GDF-15, TWSG-1 and ERFE in the bone marrow is also a limitation of the paper.
Lines 331 -333: The ERFE-mediated suppression in hepcidin that the authors refer to is not clearly shown as per data in Fig 4.

Conclusion:
Lines 352 – 358: The description in these lines is not entirely supported by the data presented in the paper. They may be considered speculations based on data obtained. The authors are encouraged to call it as such and maybe provide a summary figure to describe it better. This must not be included in the conclusion section.
Lines 358 to 361 accurately state the conclusions that can be drawn from this study and can be retained.

---

## Round 0.2 · Major Revisions

There are several points unclarified.

Reviewer 1 ·

Basic reporting

No comment.

Experimental design

The authors have addressed the key points raised in my review. I have a minor issue with the way qPCR data are presented. The authors show ΔCt values, where an increase means decrease in mRNA expression. This is counterintuitive for the reader. I would suggest to show normalized values relative to β-actin in wt young mice. Alternatively, they could show -ΔCt values.

Validity of the findings

No comment.

Additional comments

The overall manuscript is improved.

Reviewer 2 ·

Basic reporting

Language needs editing.

Relevance of study unclear

Several issue raised not satisfactorily addressed.

Experimental design

The observations do not seem to contribute to knowledge in the field; it is unclear how they.are relevant.

Validity of the findings

Relevance of the observations unclear.

Additional comments

The authors have revised the manuscript to rectify very fundamental defects in their original submission. They report various observations. However, how their observations are relevant or add to knowledge in the field are unclear. Why would it matter if “Iron homeostasis in a mouse model of thalassemia intermedia is altered between adolescence and adulthood”? The authors state that their study “underscores the importance of the age of thalassemic mice in the study of molecular or pathophysiological changes under thalassemic condition”. It is hard to understand how this is relevant. They have not studied many key parameters, such as serum levels of hepcidin or expression of erythroid regulators involved, which may have been added something to the new and different to the study. In addition, they have addressed many of the reviewer’s comments in a rather lazy way, and not very meticulously. Too many of the issues raised have merely been mentioned in the manuscript as limitations. I do not think the manuscript adds anything significant to the field.

English language editing is required.

Reviewer 3 ·

Basic reporting

N/A

Experimental design

N/A

Validity of the findings

N/A

Additional comments

It is well-appreciated that the authors have addressed most of the comments and re-analyzed the data by 2-way ANOVA which I think is the correct statistical test to be used for the 2 x 2 design of this study.
Some points to note and address in the revised manuscript are listed below:

Important
1. Whenever a significant interaction is present (in 2-way ANOVA) post-hoc tests should be done for ALL the relevant pair-wise comparisons. For example, in table 2, the superscript “a” and “b” have been used for comparisons between adolescent and adult mice within phenotypes. Although the results section state that values for Hb etc were significantly higher in WT than BKO for both adult and adolescent mice, this fact is not shown in the table. The authors are encouraged to used additional superscripts to show significant differences between WT adolescent and adult mice and similarly for BKO adolescent and adult mice. This will give a clearer picture. Same applies for liver iron, spleen iron etc in Table 2 and in Table 4 and also for all the figures.
2. The figure legends shown just above each figure in the pdf provided for review states that the statistical test used was 1-way ANOVA. I suppose this is from the original submission and must be deleted. The list of figure legends given after references is correct.
3. The authors have now shown the delta Ct values in the figures instead of fold changes and this is very good. However, the fact that delta Ct (and not minus delta Ct) was shown makes interpretation a little complicated because a decrease in value indicates induction and an increase in value indicates repression. To overcome this, I suggest minus Ct be shown on the y-axis so that an increase will indicate induction and a decrease will indicate repression. This will make interpretation a little easier and straight-forward for the reader.

Minor:
Discussion:
1. It is interesting that in adult BKO, hepcidin mRNA increased compared to adolescent but the hepcidin-liver iron ratio did not change, suggesting that the increase in hepcidin was appropriate for the increase in liver iron in these mice. Recently it has been shown that ERFE acts by binding to BMP6 (Arezes J et al Blood 2018). This can be added to the discussion.
2. The increase in mRNA expression of duodenal iron transporters is not explained by the effects of hepcidin because these changes were seen in adolescent mice when hepcidin was similar between WT and BKO mice. Perhaps the effect of hypoxia (due to anemia) on stability of duodenal HIF2 (which is known to regulate dcytb, dmt1 and ferroportin mRNA levels) may be involved. This can be discussed.
3. It is good practise to submit the ARRIVE checklist for all animal-based studies. I would strongly suggest that this checklist be submitted with the revised manuscript as a supplementary file.

---

## Round 0.3 · accepted · Accept

We are pleased to inform you that your revision has been accepted for publication in PeerJ.

Looking forward to reading your article.

Thanks a lot

Sincerely,
Editor
Cheorl-Ho